# The Role of the Planetary Diet in Managing Metabolic Syndrome and Cardiovascular Disease: A Narrative Review

**DOI:** 10.3390/nu17050862

**Published:** 2025-02-28

**Authors:** Aleksandra Muszalska, Julia Wiecanowska, Joanna Michałowska, Katarzyna Magdalena Pastusiak-Zgolińska, Izabela Polok, Kinga Łompieś, Paweł Bogdański

**Affiliations:** 1Student Scientific Club of Clinical Dietetics, Department of the Treatment of Obesity and Metabolic Disorders and of Clinical Dietetics, Poznań University of Medical Sciences, 60-355 Poznań, Poland; 2Department of the Treatment of Obesity and Metabolic Disorders and of Clinical Dietetics, Poznań University of Medical Sciences, 60-355 Poznań, Poland

**Keywords:** planetary diet, metabolic syndrome, MetS, abdominal obesity, cardiovascular disease

## Abstract

**Introduction:** The planetary health diet, introduced by the EAT–Lancet Commission, aims to address global health and sustainability challenges by promoting a plant-based diet with reduced consumption of animal-sourced foods. This diet not only contributes to environmental sustainability but also offers significant health benefits, including prevention and management of abdominal obesity, carbohydrate metabolism disorders, dyslipidemia, and elevated blood pressure. These metabolic disorders are components of metabolic syndrome, a cluster of risk factors for cardiovascular disease. **Objectives:** This narrative review aims to gather the most recent findings on the impact of the planetary diet on individual components of metabolic syndrome and on the prevention and treatment of cardiovascular disease. **Methods:** The available research on the topic was identified via searches in PubMed, Scopus, and Google Scholar. **Results:** Abdominal obesity, a major risk factor for a range of chronic diseases, can be effectively mitigated by adhering to plant-based dietary patterns, which have been shown to reduce waist circumference and improve overall cardiometabolic health. Furthermore, the planetary diet plays a crucial role in reducing the risk of type-2 diabetes and improving glycemic control, with specific dietary components such as whole grains and fiber, demonstrating positive effects on blood glucose levels. This diet is additionally associated with favorable lipid profiles, including lower levels of LDL cholesterol and total cholesterol, which are critical in the prevention of atherosclerosis and cardiovascular diseases. **Conclusions:** These multiple benefits highlight that the planetary diet may be an effective strategy for managing and preventing metabolic syndrome and cardiovascular disease. However, further research is needed to confirm its long-term efficacy and applicability across diverse populations.

## 1. Introduction

Systemic changes cannot keep up with the challenges to our approach to nutrition posed by the contemporary world. In the daily bustle, we often overlook what is on our plates. People more frequently opt for ready-made, highly processed, and calorie-dense products [1]. Research has confirmed our strong inclination towards meat, including red meat. In 2018, the average global consumption of unprocessed red meat per person amounted to 51 g per day [2]. The negative effects of its consumption are well recognized. Merely 50 g of red meat per day significantly increases mortality rates due to oncological and cardiovascular causes and type-2 diabetes (T2D) [3]. This observation is also troubling because the production of red meat, particularly beef, emits significant quantities of CO_2_ through ruminant enteric fermentation [4]. Data show that animal-derived products contribute to 18% of global greenhouse gas emissions, surpassing those from industry (16%), transportation (13.5%), and energy production (13%) [5]. The association between air pollution and the development of respiratory diseases, such as lung cancer, chronic obstructive pulmonary disease (COPD), and asthma, has been extensively studied [6]. Interestingly, in recent years, there has been a noticeable increase in the number of research publications suggesting a direct link between long-term exposure to air pollution and increased frequency of metabolic syndrome (MetS), and particularly with two of its components: abdominal obesity and carbohydrate metabolism disorders [7,8]. Treatment of MetS involves enormous financial outlays: the average annual total treatment cost of subjects with MetS has been reported to be 1.6 times greater overall than the cost of treating those without MetS, or 1.3 times more when stratified by diabetes [9]. MetS is also associated with a 1.24 times greater risk of premature death [10]. Considering the dietary habits of the population, as well as the limited time resources available for addressing climate change and the consequences of MetS, a dietary strategy based on reducing meat, rather than completely eliminating it, appears to be the optimal solution. The planetary health diet (PHD), introduced by the EAT–Lancet Commission, aims to meet these needs by taking into account both planetary and human health. This nutritional model is based on local, unprocessed, and mostly plant-based products, and involves a low intake of saturated fats and an adequate intake of fiber. It can thus be expected to have favorable effects on human health. At the same time, by limiting the consumption of meat and highly processed products, it also supports sustainability and environmental health [11].

This article aims to gather the most recent findings on the impact of the planetary diet on the individual components of MetS and cardiovascular disease (CVD) risk. The study’s primary objective is to determine whether the PHD dietary model influences the frequency and level of control of MetS components, including abdominal obesity, hypertension, glucose and lipoprotein levels, as well as CVD incidents. A secondary objective was to investigate whether specific dietary assumptions within the PHD model affect the management of individual components of the MetS.

## 2. Methods

The research for this narrative review was conducted by searching the PubMed, Scopus, and Google Scholar databases for relevant studies. The following search strategy was used to identify the relevant articles: ((((“planetary health diet”) OR (“EAT Lancet”)) OR (“planetary diet”)) OR (“sustainable diet”)) AND (((((((((((((“metabolic syndrome”) OR (“abdominal obesity”)) OR (“abdominal obesity”)) OR (hypertension)) OR (“type 2 diabetes”)) OR (diabetes)) OR (“carbohydrate disorders”)) OR (dyslipidemia)) OR (“lipid disorder”)) OR (hyperlipidemia)) OR (hypertriglyceridemia)) OR (“HDL cholesterol”)) OR (“Elevated fasting glucose”)). All the resulting abstracts were screened to identify relevant studies. Additionally, a manual search was performed to answer secondary research questions. Articles investigating underaged populations or pregnant women were excluded. The Scale for Assessment of Narrative Review Articles (SANRA) checklist was used to ensure the quality of the work [12]. Assessment performed by the authors and the final SANRA score is available as Appendix A. 

## 3. Planetary Diet

The concept of the “planetary diet” was initially introduced by EAT and presented in a report entitled “Food in the Anthropocene: The EAT–Lancet Commission on Healthy Diets from Sustainable Food Systems”, published in The Lancet on 16 June 2019 as a result of deep concern for the global food system. Without any urgent changes, future generations may live on a heavily degraded planet and suffer from undernourishment. The EAT Commission pointed out two major factors of the system impacting population health: healthy diets and sustainable production. The “win–win” scenario for both the planet and human health is a diet high in plant-based foods with lower intake of animal-sourced foods [13]. Considering the entire range of dietary patterns in terms of their effects on health and sustainability, the PHD would seem to be an optimum choice [14]. Global food production is the leading cause of environmental degradation and is a significant threat to climate stability [15]. The commission proposed strategies to cease breaching planetary boundaries, such as advocating for global and national commitments to the transition toward healthier diets, shifting agricultural priorities from focusing on high food quantity to prioritizing the production of nutritious food, enhancing food production sustainably to increase high-quality yields, implementing strong and coordinated governance for land and ocean resources, and reducing food loss and waste by at least a half [13,14]. The overall guideline for composing a planetary healthy dish involves roughly half a plate of vegetables and fruits, with the other half including whole grains, plant-based protein sources, unsaturated plant oils, and optionally small portions of animal-based protein [13]. The EAT–Lancet Commission suggests limiting the consumption of highly processed foods and encourages the choice of products that are minimally processed, plant-based, lower in saturated fat, and high in fiber [13]. The suggested daily intake of various food groups and their caloric value is presented in Table 1. A caloric intake of 2500 kcal has been used as a reference value. Table 2 gathers the key dietary recommendations of the EAT–Lancet Commission.

Following these recommendations involves high intakes of various favorable nutrients, including fiber, unsaturated fatty acids, minerals, vitamins, bioflavonoids, and antioxidants [13]. However, some concerns have emerged regarding the adequacy of the intake of specific nutrients, particularly vitamin B_12_, iron, calcium and zinc. Insufficient intake of these compounds may occur because of the limited consumption of animal-based foods, which are rich in them, and their limited bioavailability from plant sources [11]. A number of strategies have been suggested to fill these micronutrient gaps, including fortification, supplementation, inclusion of indigenous foods and practices, and increasing dietary awareness and knowledge—all of which can assist in composing well-balanced meals with higher micronutrient bioavailability [16].

## 4. Metabolic Syndrome

MetS does not constitute a separate disease entity, but rather denotes the concurrent presence of specific risk factors for atherosclerosis or T2D and cardiovascular complications. These include the combination of interdependent metabolic disturbances: insulin resistance, visceral adiposity, dyslipidemia, and hypertension [17]. According to a meta-analysis performed by Mottillo et al., MetS is associated with a doubling of cardiovascular outcomes, including CVD, CVD mortality, and myocardial infarction (MI) [18]. Noubiap et al. estimated that the global prevalence of MetS ranges from just under 13% to 31.4%, depending on the diagnostic criteria considered. Researchers have shown that residents of the eastern Mediterranean and the Americas are diagnosed with MetS significantly more frequently than people from the rest of the world [19]. The diagnostic criteria for MetS have changed several times since they were first described in 1998 by the World Health Organization (WHO) [20]. For the past sixteen years, the most widely used definition has been proposed in the joint interim statement of the International Diabetes Federation (IDF), the National Heart, Lung, and Blood Institute (NHLBI), the American Heart Association (AHA), the World Heart Federation (WHF), the International Atherosclerosis Society (IAS), and the International Association for the Study of Obesity (IASO). According to these 2009 criteria, MetS is identified when three or more of the following five criteria are met: elevated waist circumference (WC), blood pressure exceeding 130/85 mmHg (or antihypertensive treatment), fasting triglyceride (TG) concentration equal or above 150 mg/dL (or treatment for elevated TG), fasting high-density lipoprotein (HDL) level below 40 mg/dL for men or 50 mg/dL for women (or treatment for reduced HDL), and fasting blood glucose levels ≥ 100 mg/dL (or antidiabetic treatment). It should be noted that the WC cut-offs differ depending on population- and country-specific definitions. For Caucasian populations, the recommended WC threshold for abdominal obesity is 80 or 88 cm for women and 94 or 102 cm for men, depending on the guidelines. For Asian populations, these cut-offs are usually slightly lower [21]. Diagnosing MetS is relatively inexpensive [9]. A primary care physician can use a patient interview, measurements of abdominal circumference and a blood pressure measurement taken in the office to raise suspicion of the presence of MetS with high likelihood. The appropriate blood tests—fasting glucose and lipid profile—can then be ordered to complete the diagnosis [22]. The key to assessing the risk factors for MetS is the individual’s lifestyle. Individuals with a sedentary lifestyle, who smoke cigarettes, neglect sleep hygiene, and consume a diet high in saturated fats are significantly more susceptible to developing MetS and CVD [23].

## 5. The Planetary Diet and Obesity

Undernutrition persists in the modern world alongside a growing incidence of overweight and obesity, with poor dietary habits playing an important role in both of those states. The EAT–Lancet summary underlines the significance of this problem, as current data suggest that 2.1 billion adults are classified as overweight or obese [13]. Abdominal obesity (AO) has significantly increased over the years. Current research findings show that even individuals with body mass indices (BMI) below 20 kg/m^2^ can still experience AO and related comorbidities. AO, the accumulation of excess fat in the abdominal area, is recognized as a form of obesity with significant health risks [24]. The latest guidelines define AO as waist circumference ≥ 102 cm or 94 cm in men and ≥88 cm or 80 cm in women (or in Asians WC ≥ 90 cm in men and ≥80 cm in women); waist-hip ratio (WHR) > 0.85 in women and WHR > 0.9 in men [25]. AO serves as a marker for the buildup of triacylglycerols in the liver and muscles and is a risk factor for diseases such as CVD, hypertension, T2D, cancer, kidney disorders, and nonalcoholic fatty liver disease (NAFLD). Research shows that mesenteric fat tissue exhibits higher lipogenic activity than subcutaneous fat and other body regions, particularly in the context of a high-calorie diet and a sedentary lifestyle [24].

### 5.1. The Planetary Diet in the Prevention and Treatment of Obesity

Plant-based diets have consistently been linked to positive cardiometabolic outcomes, including a reduced risk of developing MetS and its components, such as obesity. Moreover, these diets have been demonstrated to lower all-cause mortality and reduce the risk of obesity. Diet plays a key role in preventing weight gain. Limiting refined grains, sugar, and saturated fats, while increasing the intake of whole grains, vegetables, nuts, fruits, and legumes, has been associated with a lower risk of overweight and obesity [26,27]. Various studies have shown that following a healthy, sustainable diet is associated with a decreased risk of excessive body mass, and these results have been confirmed by Reger et al. in their meta-analysis [28]. Analysis of the data from the PERSIAN cohort study revealed that subjects in the higher planetary health diet index (PHDI) quartile had a lower prevalence of MetS and abdominal obesity [29]. Moreover, an analysis of baseline data from over fourteen thousand participants in the Brazilian Longitudinal Study of Adult Health (ELSA-Brazil) showed an inverse association between adherence to PHD and obesity outcomes. Subjects with better adherence to the EAT–Lancet recommendations had lower BMI and WC, and were also 24% less likely to be overweight or obese [30]. A similar analysis was performed on a Polish population by Ambroży et al., who found that the 39.4% of 216 studied individuals who followed the PHD were less likely to have excessive weight than the nonadherents [31]. In a large prospective cohort of British adults (the 46,069 participants of the EPIC–Oxford study), high adherence to the EAT–Lancet score was associated with an approximately 1.4 kg/m^2^ lower BMI [32]. Klapp et al. conducted a study on a multiethnic cohort of African Americans, Japanese Americans, European Americans, Native Hawaiians and Latinos, in which anthropometric data and dietary intake were assessed and analyzed with the EAT–Lancet diet score. The study found that higher EAT–Lancet diet scores were associated with lower risks of obesity (Hazard Ratio [HR]: 0.76; 95% CI: 0.73, 0.79 for tertile 3 vs. 1) [33]. Similar results were observed by Zhan et al. in the National Health and Nutrition Examination Survey (NHANES) cohort. Subjects in the highest PHDI quintile had lower prevalence ratios of obesity (0.59; 95% CI: 0.50, 0.69) and abdominal obesity (0.74; 95% CI: 0.66, 0.82) than the lowest quintile [34]. On the other hand, Langmann et. al. did not find an association between adherence to the EAT–Lancet diet and weight upon a five-year follow-up. The study population included over 44,000 participants from the Danish Diet, Cancer, and Health cohort. At baseline, the weight, WC, and BMI were lower in subjects with a high planetary diet score than in those with lower scores. At the five-year follow-up, greater adherence to the EAT–Lancet diet was associated with a lower risk of obesity (Relative Risk (RR): 0.89, 95% CI: 0.82, 0.98) and elevated WC (RR: 0.95, 95% CI: 0.93, 0.96). Interestingly, a higher planetary diet score was not associated with follow-up weight when adjusted for baseline weight and confounders. The authors concluded that higher adherence to the EAT–Lancet planetary diet does not contribute to development of obesity [35]. It should be noted that conflicts in these data may be associated with the different scoring methods used to assess adherence to the planetary diet.

Studies investigating plant-based dietary patterns have proven beneficial effects on body weight and waist circumference. According to the meta-analysis of eleven trials performed by Wang et al., patients who follow a vegetarian diet experience greater weight loss than omnivores (−2.88 kg, *p* < 0.001) [36]. Similar findings have been presented by Huang et al. in their meta-analysis of randomized controlled trials (RCTs) conducted to compare body weight changes between individuals following vegetarian diets and those on nonvegetarian diets. It was found that patients on vegetarian diets experienced an average weight loss of about two kilograms more than those in the control group (mean difference −2.02 kg, *p* < 0.001) [37]. To our knowledge, no randomized studies investigating the effectiveness of PHD in the treatment of obesity have ever been carried out. However, in the first longitudinal study, Suikki et al. did not find any significant associations between overall diet quality (as measured by the Planetary Health Diet Score and the Recommended Finnish Diet Score) and anthropometric changes in a Finnish population. It should be noted that Suikki et al. suggested that this result may be at least partially explained by the low adherence to the recommendations in the population [38]. Further studies are needed to assess the potential usefulness of the PHD in obesity treatment.

### 5.2. The Impact of Individual Principles of the Planetary Diet on Obesity

Various food groups and components present in the PHD may modify the risk of obesity and its development. In the PREDIMED study trial of an elderly population at high risk of CVD, it was observed that increased consumption of certain ultra-processed foods—including snacks, fast food, preprepared dishes, processed meats, and sweets—was associated with changes in WC and primarily associated with weight gain [39]. This link has also been confirmed by data from the PREDIMED-Plus study, which showed that the consumption of highly processed foods by older adults is associated with higher body weight and WC, as well as complications of obesity (hypertension, elevated blood glucose, and dyslipidemia). Moreover, alcohol consumption was associated with a greater increase in both weight and WC [40]. The planetary diet, whose dietary recommendations suggest a greater intake of low-fat dairy products, was associated with less weight gain, while vegetables and nuts were associated with a smaller WC increase. Furthermore, this study reinforces earlier evidence on the significance of carbohydrate quality in preventing obesity. Foods that are low in fiber but rich in refined carbohydrates or starches are linked to greater weight gain [39]. Additionally, the fat profile of the PHD is predominantly composed of monounsaturated (MUFA) and polyunsaturated fatty acids (PUFA), with lower saturated fatty acid (SFA) intake than other dietary patterns [13]. The replacement of SFA with MUFA and PUFA has been associated with anti-inflammatory benefits [41,42]. The antioxidant properties of the nutrients and bioactive compounds found in plant-based foods, such as vitamins C and E, β-carotene, and polyphenols, have been associated with the prevention of MetS and its components, such as obesity [26]. Dietary fiber has been associated with body mass, and its beneficial mechanisms include increased satiety due to greater food volume, lower glycemic and insulinemic responses to meals, and enhancements in gut microbiome health [43]. The PREDIMED study examined the inverse relationship between vegetable fat intake and waist circumference. Nuts, despite their high unsaturated fat content, are rich in fiber, bioactive compounds, vegetable protein, and fatty acids, and may exert anti-obesity effects by boosting thermogenesis, resting energy expenditure, and altering fat oxidation [41]. Another study suggested that increasing the intake of vegetable fats from natural sources such as nuts and extra virgin olive oil can significantly contribute to weight reduction [44]. More importantly, the fat composition of food can affect not only body mass, but also the distribution of fat tissue in the human body. In their overfeeding protocol, Rosqvist et al. studied the influence of surplus PUFA (sunflower oil) vs. SFA (palm oil) intake. Magnetic resonance imaging (MRI) scans showed that, despite the comparable weight gains between the two groups, excessive consumption of palm oil was associated with a greater increase in the amount of abdominal and hepatic fat [45].

## 6. The Planetary Diet and Carbohydrate Metabolism Disorders

Prediabetes is a precursor of diabetes and often progresses to the full condition if not managed effectively. The diagnosis of both prediabetes and diabetes relies on glucose measurements, typically using fasting plasma glucose (FPG) tests, oral glucose tolerance tests (OGTT), or glycated hemoglobin (HbA1C) levels [46]. Prediabetes is characterized by FPG levels of 100–125 mg/dL (or 110–125 mg/dL, depending on the guidelines), two-hour post-OGTT glucose level of 140–199 mg/dL, or HbA1C of 5.7–6.4%. In turn, diabetes is diagnosed by FPG ≥ 126 mg/dL, two-hour post-OGTT glucose level ≥ 200 mg/dL, or HbA1C ≥ 6.5%. Diabetes may also be diagnosed on the basis of a random plasma glucose level equal or higher than 200 mg/dL in individuals presenting classical symptoms of hyperglycemia or a hyperglycemic crisis [47,48]. Hyperglycemia and insulin resistance in diabetes and prediabetes lead to increased reactive oxygen species, initiating intracellular molecular signaling. This process promotes a prothrombotic state and heightens inflammatory mediators, accelerating atherosclerosis and macrovascular complications. Individuals with diabetes and prediabetes consequently face greater risks of CVD and cardiovascular events, including myocardial infarction, stroke, and peripheral artery disease [48,49].

### 6.1. The Planetary Diet in the Prevention and Treatment of Carbohydrate Metabolism Disorders

Current research findings suggest that adherence to the EAT–Lancet reference diet is associated with a reduced risk of developing diabetes [15,32,50,51,52]. For instance, Knuppel et al. have demonstrated that high adherence to this diet correlates with a 59% reduction in diabetes risk [32]. Similarly, López et al. conducted a prospective cohort study with 74,671 Mexican women, finding that higher adherence to the EAT–Lancet healthy reference diet (EAT-HRD) was associated with a lower incidence of T2D (HR: 0.90; 95% CI: 0.75, 1.10). Specifically, adherence to the recommendations on red meat (HR: 0.79; 95% CI: 0.63, 0.99), legumes (HR: 0.92; 95% CI: 0.84, 0.99), and fish (HR: 0.92; 95% CI: 0.85, 1.00) were associated with a reduced incidence of T2D in comparison to those who did not follow these guidelines. It should be noted that the differences were just at the limit of statistical significance or else were not statistically significant. Interestingly, adherence to the EAT-HRD recommendations for dairy (HR: 1.12; 95% CI: 1.04, 1.21) and added sugars (HR: 1.11; 95% CI: 1.02, 1.21) were linked to an increased incidence of T2D. The authors suggest that these imprecise and weak associations can be explained by various factors that include the scoring system employed, the possibility of unmeasured confounding, and great variability within the food groups. This latter limitation may play a crucial role in the association found for dairy, as low-fat and high-fat dairy have the opposite effects on T2D risk, yet were included in the same food group [53]. These methodological shortcomings have not gone unnoticed by researchers. Unger et al. shared their thoughts on the matter in a letter to the editor of the European Journal of Clinical Nutrition [54]. Additionally, in the analysis of the UK Biobank cohort, Xu et al. found that each one-point increase in the EAT–Lancet diet pattern score corresponded with a 6% reduction in the risk of T2D (HR: 0.94; 95% CI: 0.91–0.97) over a median follow-up period of ten years. Furthermore, the same research demonstrated that adherence to the recommended intake of potatoes, a variety of vegetables, fruits, dairy products, as well as beef, lamb, pork, and eggs was associated with a reduced risk of developing T2D in this analysis [55]. The recent Multiethnic Cohort (MEC) Study suggests that adherence to the EAT–Lancet diet only partially contributes to a reduction in the incidence of T2D. Subjects in the highest tertile had a lower risk of T2D than participants in the lowest tertile (HR 0.76; 95% CI 0.73, 0.79), but the authors concluded that this observation was related to the lower risk of obesity, as the association was attenuated after BMI adjustment (HR: 0.97; 95% CI: 0.94, 0.99) [33]. Similar results were obtained by Langmann et al. who aimed to assess the association between the adherence to the EAT–Lancet diet and T2D risk in the Danish Diet, Cancer and Health cohort. Almost 55,000 participants were included in the analysis, of whom 7130 developed T2D during a median fifteen-year follow-up period. Subjects with highest EAT–Lancet diet scores had a lower risk of developing diabetes than those with the lowest scores (HR: 0.78, 95% CI: 0.71, 0.86). After adjusting the results for potential mediators, including BMI, the HR increased to 0.83 (95% CI: 0.76, 0.92) [52]. Zhanga et al. examined the relationship between adherence to the EAT–Lancet diet and the risk of T2D, as well as whether this association varied based on genetic predisposition to T2D, as measured by a polygenic risk score (PRS). Participants were classed as having low (quintile 1), medium (quintiles 2–4), or high (quintile 5) genetic risk, as determined by their PRS. Over a median follow-up of 24.3 years, 4197 T2D cases (17.1%) were documented. Compared to participants with the least adherence to the EAT–Lancet diet (≤13 points), those with the highest adherence (≥23 points) had an 18% lower risk of T2D (HR = 0.82; 95% CI: 0.70–0.96; *p* for trend < 0.01). No significant gene-diet interactions were observed, whether multiplicative (*p* = 0.59) or additive (*p* = 0.44). The highest risk of T2D was noted among individuals with high genetic risk and low adherence to the diet (HR = 1.79; 95% CI: 1.63–1.96). The findings suggest that high adherence to the EAT–Lancet diet reduces the risk of T2D, regardless of genetic susceptibility [56].

As previously noted, a fundamental aspect of the PHD is a significant reduction in meat consumption. Research into the effects of predominantly plant-based diets on hyperglycemia dates back to the 1950s [57]. Nearly two decades ago, Barnard and colleagues demonstrated in a randomized clinical trial that a low-fat vegan diet reduced HbA1c levels more effectively than did a conventional diet based on the 2003 guidelines of the American Diabetes Association (ADA). In this study, the vegan diet was characterized by approximately 10% of energy from fat, 15% protein, and 75% carbohydrates, whereas the ADA diet had 15–20% protein, less than 7% saturated fat, 60–70% carbohydrate and monounsaturated fats, and cholesterol ≤ 200 mg/day. At week 22, hemoglobin A1c decreased by 1.23 points in the vegan group, in comparison to 0.38 points in the ADA group (in patients who did not change medication during the study). Moreover, body weight change correlated with HbA1c [58]. In 2014, a review and meta-analysis of controlled clinical trials on vegetarian diets for treating T2D revealed a significant reduction in HbA1c, by 0.39 percentage points compared to control diets. Six studies were included in the meta-analysis and comparators included classical low-fat, omnivorous, ADA, and diabetic diets [59]. This reduction is approximately half of that observed with metformin, a commonly used oral hypoglycemic agent. A recent retrospective cohort study reported that metformin, administered at a dose of 1000 mg, reduced HbA1c levels by 0.9 percentage points [60].

The question of whether a strictly plant-based diet can assist in managing T2D remains unresolved. This is because current studies do not allow us to pinpoint which aspects of a meatless diet contribute to improved glycemic control. It is unclear whether the benefits come from reduced animal protein intake, associated weight loss, increased dietary fiber, or a combination of these factors [57]. Another hypothesis links high red meat consumption with T2D due to its heme iron content. Epidemiological evidence shows that increased heme iron intake correlates with unfavorable plasma profiles of insulinemia, inflammation, and T2D-linked metabolites [61]. A recent systematic review using network meta-analysis (NMA) of randomized trials compared the efficacy of ten different dietary models (including vegetarian/vegan diets) in glycemic control for T2D patients. The NMA findings indicate that ketogenic, low-carbohydrate, and low-fat diets significantly reduced HbA1c (−0.73, −0.69, and −1.82, respectively), while moderate-carbohydrate, low glycemic index, Mediterranean, high-protein, and low-fat diets significantly lowered fasting glucose (−1.30, −1.26, −0.95, −0.89, and −0.75, respectively) compared to control diets. The combined outcomes clustered ranking plot suggested that ketogenic, Mediterranean, moderate-carbohydrate, and low glycemic index diets were promising for controlling HbA1c and fasting glucose. A vegetarian/vegan diet did not show statistically significant improvements in glycemic control [62].

It is important to note that the PHD differs from a strictly vegetarian or vegan diet, as it does not center solely on the elimination of animal products. Instead, the PHD emphasizes a balanced approach, integrating a variety of plant-based foods while allowing moderate amounts of sustainably sourced animal-derived products. This approach aligns with the characteristics of moderate-carbohydrate and low glycemic index diets identified in the NMA as effective for glycemic control. By promoting a diet rich in whole grains, legumes, fruits, vegetables, and healthy fats, while limiting processed and high-glycemic foods, the PHD offers the potential to manage glycemic control while also ensuring sustainability benefits.

### 6.2. The Effects of the Individual Principles of the Planetary Diet on Carbohydrate Metabolism Disorders

Evidence suggests that employing PHD as a specific, well-defined nutritional plan can positively impact the prevention and treatment of T2D. Moreover, even the implementation of individual principles of this dietary model can have beneficial effects [63,64,65,66]. In their RCTs, Åberg et al. demonstrated that the consumption of whole grains significantly reduces glycemia levels, as measured by 24-h glucose monitoring systems, with a mean reduction of −0.16 mmol/L (95% CI −0.25 to −0.06). Additionally, there was a notable decrease in the mean amplitude of glycemic excursions, showing a reduction of −0.36 (95% CI −0.65 to −0.08) in individuals with T2D [63].

A systematic review and meta-analysis by Schwingshackl et al. provide clear evidence of reduced risk for developing T2D with dietary modifications that align with the principles of the PHD. The study compared the effects of twelve food groups on T2D risk and found that increased consumption of whole grains, fruits, and dairy significantly lowered the risk. Optimal consumption of these foods (two servings/day of whole grains, two to three servings/day of vegetables, two to three servings/day of fruits, and three servings/day of dairy) can lead to a 42% reduction in T2D risk. Specifically, 50 g/day of whole grains was associated with a 25% reduction in risk. Conversely, an increased intake of red meat, processed meat, and sugar-sweetened beverages (SSBs) was linked to a higher risk of T2D. Consuming two servings/day of red meat (170 g/day), four servings/day of processed meat (105 g/day), and three servings/day of SSBs (750 mL/day) can triple the risk of T2D compared to not consuming these. Avoiding these foods could reduce T2D risk by approximately 70% [64].

## 7. The Planetary Diet and Dyslipidemia

Dyslipidemias are identified by high levels of total cholesterol (TC) (more than 200 mg/dL), low-density lipoprotein cholesterol (LDL-C) (various criteria depending on cardiovascular risk), and triglycerides (TGs) (more than 150 mg/dL), along with low levels of high-density lipoprotein cholesterol (HDL-C) (less than 40 mg/dL for men and less than 50 mg/dL for women) in the blood plasma, or by various combinations of these lipid abnormalities [67,68]. Disruptions in lipid levels, which may arise from genetic factors or lifestyle choices, can contribute to the development of atherosclerosis and other cardiovascular issues. Diagnosing these conditions typically involves lipid profile tests to determine the appropriate target levels for cardiovascular health. Treatment aims to reduce risks by addressing specific lipid imbalances, promoting lifestyle changes, and taking into account additional health conditions to tailor treatment [68].

### 7.1. The Planetary Diet in Prevention and Treatment of Dyslipidemia

Individuals in the large cohort study of Cacau et al. who closely followed the EAT–Lancet diet, represented by the highest quintile of PHDI, exhibited reduced levels of LDL-C (β: 4.10; 95% CI: 5.97–2.23), TC (β: 3.15; 95% CI: 5.30–1.01), and non-HDL-C (β: 2.57; 95% CI: 4.62–0.52). Conversely, no correlation was observed between adherence to the EAT–Lancet diet and levels of TGs, HDL-C, or the homeostatic model assessment for insulin resistance (HOMA-IR) [69]. Moreover, in previously described research using data from the EPIC–Oxford study, cross-sectional analysis showed that high adherence to the EAT–Lancet score was associated with about 0.5 mmol/L lower plasma non-HDL cholesterol [32]. Studies suggest that vegan diets, which include specific amounts of plant sterols, viscous fibers, soy protein, and nuts, have a more favorable impact on lipid profile than the NCEP diets (National Cholesterol Education Program) in individuals with hypercholesterolemia [70]. In one meta-analysis, plant-based diets showed reduced total cholesterol compared to omnivorous diets, with mean differences of −0.34 mmol/L and −0.30 mmol/L for LDL-C [71], respectively.

### 7.2. The Impact of the Individual Principles of the Planetary Diet on Dyslipidemia

In 2020, Trautwein et al. published a review that compiled current knowledge of the role of specific components of a plant-based diet in managing dyslipidemia [72]. As previously mentioned, one of the key principles of the PHD is the significant reduction in the intake of animal-derived protein, which is a primary source of SFA [73]. Conversely, plant-based fats, such as vegetable oils, are typically rich in unsaturated fatty acids. These unsaturated fatty acids include monounsaturated fatty acids, such as oleic acid, as well as polyunsaturated fatty acids. Plant-based sources of PUFA predominantly provide omega-6 (n-6) fatty acids, such as linoleic acid, along with some omega-3 (n-3) fatty acids, like α-linolenic acid [72]. Research shows that replacing SFA with unsaturated fatty acids in a diet lowers LDL-C levels without affecting HDL-C or TG levels. The reduction in LDL-C is more significant when SFA is replaced by PUFA rather than by MUFA [72,74]. Adopting a predominantly plant-based diet is an effective way to meet the recommended daily intake of dietary fiber. Vegan and vegetarian diets, in particular, are rich in dietary fiber, sourced from various plant-based foods such as whole grains, seeds, legumes, vegetables, fruits, and nuts [72]. According to the European Food Safety Authority (EFSA), a daily intake of 25 g of dietary fiber is sufficient for normal bowel function in adults. However, higher intakes—above 25 g per day—are recommended to reduce the risk of coronary heart disease [75]. Research has consistently demonstrated that adequate dietary fiber intake results in positive changes in lipid profiles, thereby contributing to a lower risk of CVD [72]. A recent review highlighted that increasing dietary fiber intake significantly reduces TC levels (based on 17 trials and 1067 participants), showing a mean difference of −0.20 mmol/L, 95% CI −0.34 to −0.06) and LDL-C (mean difference of −0.14 mmol/L, 95% CI −0.22 to −0.06). Although the reduction in HDL-C was modest, it was statistically significant (mean difference of −0.03 mmol/L, 95% CI −0.06 to −0.01), and no effect was observed on TG levels [76]. Phytosterols (PS) are another group of components widely present in plant-based foods, which are the core of PHD. These naturally occurring bioactive compounds are structurally similar to cholesterol [74]. The main PS include sitosterol, campesterol, and stigmasterol, along with their saturated forms sitostanol and campestanol. These compounds are found in a wide range of plant-based foods, such as vegetable oils (especially unrefined oils), margarine made from vegetable oils, seeds, nuts, grains, legumes, vegetables, and fruits, as well as in various fortified foods and dietary supplements containing added PS [77]. In the digestive system, PS and cholesterol compete for the same absorption mechanisms. As a result, PS can lower blood cholesterol levels by reducing the amount of cholesterol absorbed [78]. The typical daily intake of PS in general diets ranges from 200 to 400 mg. However, diets focused on plant-based foods result in higher PS intake, with vegetarian or vegan diets potentially providing up to 600 mg per day [79,80]. A large meta-analysis of 124 clinical trials demonstrated that PS consumption significantly reduces LDL-C levels in a dose-dependent manner by 6–12% when intake ranges from 0.6 to 3.3 g per day, with no effect on HDL-C [81]. Another meta-analysis also indicated that PS intake lowers the levels of atherogenic apolipoproteins, such as apo-B and apo-E, while increasing levels of anti-atherogenic apolipoproteins, including apo-AI and apo-CII [82]. Although there is a lack of RCTs examining the long-term effects of PS intake on CVD outcomes, such as cardiovascular events, the established LDL-C lowering effect of PS suggests they may contribute to a reduced risk of CVD [72].

## 8. The Planetary Diet and Hypertension

Hypertension is one of the leading and most modifiable risk factors for CVD and mortality. According to the Framingham Heart Study, almost everyone is at risk of suffering from hypertension as they age. Several factors contribute to abnormal blood pressure, including obesity, high fat and alcohol consumption, excessive sodium intake and sedentary lifestyle. Additionally, a diet lacking fruits, vegetables, and whole grains has been associated with an increased risk of hypertension [83]. Due to the diverse nature of hypertension, determining the exact role of dietary factors in its development is complex; defining the most effective dietary changes for managing hypertensive patients is even more challenging [84]. Research findings suggest that nutritional interventions play a very important role in the development and management of arterial hypertension. Sodium, potassium, calcium, and magnesium all affect blood pressure levels differently, which has drawn attention to the PHD approach to hypertension treatment [85].

### 8.1. The Planetary Diet in Prevention and Treatment of Hypertension

Obesity is widely recognized as a key risk factor for elevated systolic and diastolic blood pressure and, according to research findings, weight loss can lead to meaningful blood pressure reductions. Additional benefits can be achieved by decreasing overall fat intake and improving the balance of polyunsaturated fats to saturated fats. Furthermore, these dietary fat adjustments not only lower blood pressure, but also reduce cardiovascular risk [26,83]. The meta-analysis of Lee et al. showed that vegetarian diets are associated with significant reductions in both systolic and diastolic blood pressure, as compared to an omnivorous diet [86]. In the DASH-Sodium trial (Dietary Approaches to Stop Hypertension), hypertensive and normotensive patients were recommended to follow either a standard American diet or the DASH diet, which is high in fruits, vegetables, and low-fat dairy products. Participants were later divided into groups of three levels of salt intake: approximately 8 g, 6 g, and 4 g per day. Reducing salt intake lowered blood pressure in both hypertensive and normotensive individuals. Moreover, when salt intake was reduced from 8 g/day to 4 g/day with the standard diet, blood pressure dropped by an average of 8.7/4.5 mmHg in hypertensive and 5.3/2.6 mmHg in normotensive individuals. The combined effect of the DASH diet and salt reduction resulted in blood pressure reductions of 11.5/5.7 mmHg in hypertensive and 7.1/3.7 mm Hg in normotensive participants [87]. The PHD does not contain a high sodium content, as it involves reduced quantities of processed foods. This may have beneficial effects in preventing hypertension among normotensive patients. A cohort study in China of over 11,000 participants studied the association between adherence to the EAT–Lancet diet and the risk of hypertension. Over eighteen years, hypertension occurred in 35% of participants. People with higher adherence to the PHD had a significantly lower risk of hypertension (*p* < 0.001). The connection was demonstrated in isolated systolic, isolated diastolic, and combined systolic–diastolic hypertension [88]. A similar association has been noted among European adolescents, where individuals following the EAT–Lancet diet had a lower probability of developing high blood pressure [89]. On the other hand, Shojaei et. al. did not find any association between high blood pressure and greater adherence to the planetary diet in their analysis of the data from the PERSIAN cohort study [29].

The DASH diet has gained widespread recognition as a dietary strategy for managing blood pressure. In this diet, patients are encouraged to have a high intake of fruits, vegetables, fiber, and low-fat dairy products with a reduction in sodium intake. Unprocessed foods, reduction in animal products, with a focus on vegetable and fruit consumption, are also key aspects of the PHD [13,90,91]. Filippou et al. conducted a meta-analysis and confirmed that the BP-reducing effect of the DASH diet was observed across patients from normotensive to stage 1 hypertensive, with a greater absolute reduction in SBP and DBP among hypertensive patients without antihypertensive medication [92]. In the research findings, the low sodium intake in the dietary plan seems to be the key point in the treatment of hypertensive patients [92,93]. In another study, the median change in mean arterial pressure between high-sodium and low-sodium diets was 4 mm Hg (*p* <  0.001), with an effectiveness of low-sodium diet in 73.4% of participants compared to the high-sodium diet [94]. Results from the ELSA-Brasil cohort study using data from 14,155 participants suggest that higher adherence to the EAT–Lancet diet is associated with lower levels of blood pressure. In the fully adjusted linear regression models, those individuals with higher adherence to the recommendations of PHD (5th quintile) had lower SBP (−0.81 mmHg; 95% CI: −1.54, −0.07), and DBP (−0.66 mmHg; 95% CI: −1.19, −0.12) [69]. Further studies are needed to explain the role of the EAT–Lancet diet in the treatment of hypertension.

### 8.2. The Impact of Individual Principles of the Planetary Diet on Hypertension

According to WHO recommendations, adults over sixteen years should limit their sodium intake to less than 2000 mg per day, which is equivalent to less than 5 g of salt per day [95]. The PHD naturally encourages the consumption of unprocessed, low-sodium foods such as fruits and vegetables with a high intake of vitamins, minerals, and fiber. Wang et al. conducted a meta-analysis whose findings suggest that a greater intake of ultra-processed foods notably elevates the risk of developing hypertension in adults (*p* = 0.034). Lower levels of consumption of sugar-sweetened beverages, a lower risk of obesity, low levels of saturated and trans-fatty acids, and high amounts of fiber and micronutrients were found to contribute to the lower risk of high blood pressure in that study [96]. The key factors of PHD assist in managing inflammation and reducing oxidative stress and, in general, reduce the risk of hypertension and CVD. With a smaller incidence of dyslipidemia, atherosclerosis, obesity, insulin resistance and T2D, patients following the EAT–Lancet diet had a statistically significant 8% reduction in the risk of mortality, and a 16.1% reduction in the risk of CVD [97]. It has been discussed whether the EAT–Lancet diet provides all necessary nutrients, such as calcium and iron [11]. A study investigating the correlation between adherence to the PHDI and nutrient adequacy found that the PHD improves the adequacy of fiber and potassium [98]. Increasing potassium intake has been shown to reduce blood pressure in hypertensive adults, particularly in individuals who are Black, older, or have a high dietary sodium intake, and supplementing potassium is used in the pharmacological treatment of hypertension [99,100]. Evidence collected from the Brazilian study shows a negative correlation between hypertension and the EAT–Lancet Diet. While emphasizing the role of the appropriate intake of fruits, vegetables, and legumes, limiting red processed meat, trans-fats, saturated fats, and ultra processed foods was associated with lower systolic blood pressure (β −0.14, 95% CI −0.25 to −0.01; *p* <  0.05) [101]. Furthermore, Frank et al. conducted a study to compare PHDI, the Healthy Eating Index-2015 (HEI-2015), and DASH and to examine their relation to cardiometabolic health, finding that the better the diet quality, the lower the probability of cardiometabolic risk. All three indices were connected to a reduction in blood pressure by 2.9 percentage points for PHDI and 3.9 percentage points for DASH. Although the PHDI had a weaker association with lower blood pressure than the other dietary patterns, it was the only dietary approach that considered both human health and environmental issues [102].

## 9. The Planetary Diet and Overall Cardiovascular Disease Risk

In previous sections, we have shown that the PHD can significantly contribute to improving control of the components of MetS. It is consequently necessary to investigate the potential of the PHD in mitigating overall cardiovascular risk, which is measured using a variety of endpoints. The data from PREDIMED-Plus show that higher adherence to the pro-vegetarian food patterns (based on higher consumption of vegetables, fruits, legumes, grains, potatoes, nuts, and olive oil and lower intake of meat and meat products, animal fats, eggs, diary, fish, and seafood, as well as fruit juice, sweetened beverages and sweets) was associated with lower levels of the MetS components in the elderly. It was also associated with overall cardiometabolic risk [103]. One study on the topic was an extensive prospective cohort study in the USA, which included 62,919 women from the Nurses’ Health Study (NHS) I, 88,535 women from the NHS II, and 42,164 men from the Health Professionals Follow-up Study (HPFS). The authors calculated the PHDI based on fifteen food groups. Whole grains, vegetables, fruit, fish and shellfish, nuts, seeds, non-soy legumes, soy foods, and unsaturated oils were scored positively, whereas starchy vegetables, dairy, red or processed meat, poultry, eggs, saturated fats and trans-fats, and added sugar received negative scores. Scores for each food group were summed to yield a total of 0–140. In the context of a multivariable-adjusted meta-analysis, it was observed that participants in the highest quintile of PHDI scores exhibited a significantly lower risk of developing incident CVD than those in the lowest quintile (HR: 0.83; 95% CI: 0.78, 0.89). Furthermore, when analyzing specific subtypes of CVD, individuals in the highest quintile of PHDI were also found to have a reduced risk of coronary heart disease (HR:0.81; 95% CI: 0.74, 0.88) and total stroke (HR: 0.86; 95% CI: 0.78, 0.95) [104]. Another study based on similar data included 66,692 healthy females from the NHS (1986–2019), 92,438 females from the NHS II (1989–2019), and 47,274 males from the HPFS (1986–2018). The PHDI was calculated using data from a semiquantitative food frequency questionnaire. Higher scores were associated with a lower risk of death from CVD (HR: 0.86; 95% CI: 0.81, 0.91) [105]. A prospective cohort study from the UK included 114,165 participants from the UK Biobank who completed at least two 24-h dietary recalls and were initially free of CVD. The participants with the highest adherence to PHD had a lower risk of total CVD (HR: 0.79; 95% CI: 0.74, 0.84), ischemic heart disease (HR: 0.73; 95% CI: 0.67, 0.79), atrial fibrillation (HR: 0.90; 95% CI: 0.82, 0.99), heart failure (HR: 0.69; 95% CI: 0.59, 0.82), and stroke (HR: 0.88; 95% CI: 0.75, 1.04), in comparison to the least adherent participants. Moreover, participants with a high genetic risk and low PHD scores had a 48% higher risk of CVD. The population-attributable risk of CVD for poor adherence to PHD ranged from 8.79% to 14.00% [106]. Moreover, another analysis of UK Biobank data showed that the multivariable-adjusted hazard ratio for the highest (89.9–128.5 points) vs. the lowest quartile (21.1–71.1 points) of PHDI adherence was 0.86 (95% CI: 0.79, 0.94) for CVD, 0.88 (95% CI: 0.80, 0.97) for myocardial infarction, and 0.82 (95% CI: 0.70, 0.97) for stroke. The association was linear until 80 points of adherence to PHDI [107]. A Dutch prospective study evaluated compliance to the PHD using an original method and showed that high adherence was associated with a 14% lower risk of CVD, 12% lower risk of coronary heart disease, and 11% lower risk of stroke [108]. Similar results were obtained in an analysis of data from the Malmö Diet and Cancer cohort. Zgang et al. found that the hazard ratio for coronary events among participants who had the greatest adherence to the EAT–Lancet diet in comparison to the lowest adherence was 0.8 (95% CI: 0.67, 0.96) [109]. However, not all studies confirm these results. In the Swiss Cohort from the CoLaus/PsyCoLaus cohort study, no significant association was found between the PHD adherence and the risk of cardiovascular events. Concurrently, a negative association was found for all-cause mortality, leading the author to conclude that high adherence to the PHD is associated with a potential 30% lower risk of overall mortality [110]. In a Danish study, adherence to the EAT–Lancet diet was associated with a lower risk of stroke, but this association was not statistically significant [111]. On the other hand, significant associations were found for atrial fibrillation and heart failure. In both studies, higher adherence to the EAT–Lancet diet was associated with lower incidence of those disease entities [112,113]. A recent systematic review and meta-analysis, which included most of the above-mentioned studies, showed a negative association between PHD adherence and CVD mortality (HR: 0.84, 95% CI: 0.81, 0.87). Moreover, associations were found for T2D (HR: 0.78; 95% CI: 0.65, 0.92), all-cause mortality (HR: 0.83; 95% CI: 0.78, 0.89), and cancer mortality (HR: 0.86; 95% CI: 0.80, 0.92) [114].

## 10. The EAT–Lancet Planetary Health Diet: Clinical Perspective and Future Directions

The global burden of obesity and the associated cardiometabolic complications pose a great challenge to health care providers, society, and the economy. According to data from the World Health Organization, 60% of adults and nearly one in three children in Europe are overweight [115]. The global rates are lower, at around 40%, as the prevalence of excessive body mass varies by population and region. Nevertheless, it is estimated that, worldwide, one in eight people are currently suffering from obesity [116]. As abdominal obesity is the most frequently observed component of MetS, this phenomenon is directly associated with a global increase in its frequency, and consequently, in the growing prevalence of other noncommunicable diseases, including T2D, CVD, some types of cancer, and chronic respiratory diseases [115]. Comprehensive, integrated, and effective strategies are needed to support patients with metabolic disorders, and, most importantly, to prevent populations from developing them in the first place. Nonpharmacological strategies are the main pillar of maintaining good health. Moreover, the everyday lifestyle choices of each individual can significantly influence the environment. The EAT–Lancet Commission present the planetary health diet as an optimal diet for both human and planetary health. Our research confirms that following its specific recommendations is associated with improved lipid profile, reduced obesity risk, improved blood pressure, and reduced risk of T2D and CVD. Taking into account these metabolic benefits and the minimal risk of negative impact on health, it seems justified to recommend PHD for the prevention and treatment of metabolic disorders.

Nevertheless, we are aware of various limitations of the available research data. Most of the studies investigating PHD have been retrospective studies that analyzed the data collected years before the introduction of the EAT–Lancet recommendations in 2019. Moreover, inconsistent scoring systems and research methodologies may raise concerns about the reliability of the data. Future research, including prospective and interventional studies, is needed to explicitly confirm how following the PHD recommendations can affect human health.

## 11. Conclusions

The PHD demonstrates significant promise in managing MetS, including its role in addressing obesity. Its plant-based focus, richness in fiber, and low energy density supports weight reduction and helps to mitigate obesity-related risk factors. In addition, this diet improves lipid profile, enhances insulin sensitivity, and supports hypertension management through reduced sodium intake and increased potassium and fiber consumption. The inclusion of phytosterols, naturally present in plant-based foods, further contributes to cardiovascular health by significantly lowering LDL cholesterol levels and improving the balance of atherogenic and anti-atherogenic apolipoproteins. These combined benefits emphasize that the PHD may be an effective strategy for managing and preventing MetS and CVD; however, further research is needed to confirm its long-term efficacy and applicability across diverse populations.

## Figures and Tables

**Table 1 nutrients-17-00862-t001:** Summary of the EAT Commission’s recommendations on dietary choices [13].

	Type of Product	Macronutrient Intake (Range) [g/Day]	Caloric Intake [kcal/Day]
Vegetables	All vegetables	300 (200–600)	78
Fruits	All fruit	200 (100–300)	126
Whole grains	Rice, wheat, corn, and other grains	232	811
Tubers or starchy vegetables	Potatoes and cassava	50 (0–100)	39
Dairy	Whole milk or equivalents	250 (0–500)	153
Plant-based protein sources	Dry beans, lentils and peas	50 (0–100)	172
	Soy foods	25 (0–50)	112
	Peanuts	25 (0–75)	142
	Tree nuts	25	149
Animal protein sources	Beef and lamb	7 (0–14)	15
	Pork	7 (0–14)	15
	Chicken and other poultry	29 (0–58)	62
	Eggs	13 (0–25)	19
	Fish	28 (0–100)	40
Added fats	Palm oil	6.8 (0–6.8)	60
	Unsaturated oils (olive, soybean, rapeseed, sunflower, and peanut oil)	40 (20–80)	354
	Dairy fats	0	0
	Lard or tallow	5 (0–5)	36
Additional sugars	All sweeteners	31	120

**Table 2 nutrients-17-00862-t002:** Planetary diet recommendations proposed by the EAT–Lancet Commission [13].

Dietary Components	EAT Commission Recommendations
Protein sources	Reduction of red and processed meat Lower intake of dairy products, preference for low-fat dairy products Sources of omega-3 fatty acids such as fish (about 28 g per day) or plant sources of alfa-linolenic acid Intake of eggs of about 13 g per day or 1–5 per week Intake of nuts of 50 g per day Legumes: 50 g dry weight per day of beans, lentils and peas and 25 g per day of soy beans Other protein sources like insects, cyanobacterium (blue-green algae) and in-vitro meat are important in smaller populations, but were not considered in the general world consumption
Carbohydrate sources	High intake of whole grains and fiber from grain sources, while reducing refined grain and potatoes It is suggested to gain around 60% of the energy intake from complex carbohydrates, mostly whole grains 232 g per day of whole grains and 50 g per day (maximum 100 g) of tubers and starchy vegetables
Fruits and vegetables	Sources of micronutrients are important and play a key role in the prevention of cardiovascular disease, cancer, and obesity About five servings of fruit and vegetables per day or 300 g per day of vegetables and 200 g per day of fruits
Added fat	Low-fat diet to prevent weight gain and cardiovascular disease Consumption of mostly plant oils low in saturated fats, reduction of animal fats 50 g per day of total added fat (mostly unsaturated plant oils)
Sugars and sweeteners	Sugar intake should make up less than 10% of total energy intake, with a tendency towards around 5% or less

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
