# Peer review of "The Role of the Planetary Diet in Managing Metabolic Syndrome and Cardiovascular Disease: A Narrative Review"

_nutrients, 2025, doi:10.3390/nu17050862_

Round 1
Reviewer 1 Report
Comments and Suggestions for Authors
Dear Authors,
the comments in the annex file.
Best

Author Response
Dear Reviewer,
Thank you for your comments and suggestions. Please, see our reply to each question in the file attached.
Kind regards

Reviewer 2 Report
Comments and Suggestions for Authors
- The Introduction should include a short presentation of the planetary diet’s origin and main principles when the notion is first introduced (whilst keeping the current dedicated section to discuss it extensively).
- The section dedicated to the planetary diet:
- should come immediately after the Introduction section and precede the section referring to the metabolic syndrome (and all others that are disease-related);
- should also discuss the main limitations of the planetary diet – e.g., the potential risk of some micronutrient deficiencies (vitamins A and D, calcium or even sodium, etc.), insufficient evidence on nutritional overall long-term effects, etc.; the limitations of the planetary diet should also be briefly acknowledged in the Conclusions.
- The search strategy, which is only briefly mentioned at the end of the Introduction section, should be extended to accurately describe the procedures used to find and critically select their sources.
- Lines 53-55, the definition of the metabolic syndrome: the fundamental pathophysiologic basis of the metabolic syndrome is insulin resistance, which should therefore be acknowledged as such.
- Lines 57-59: since the IDF diagnostic criteria for metabolic syndrome (which the authors mention a few lines below) are currently accepted worldwide, we cannot speak anymore of wide prevalence variations due to “diagnostic criteria adopted in each country”.
- Lines 75-76: waist circumference thresholds for the metabolic syndrome are currently unique and worldwide accepted as 80 cm in women and 94 cm in men of Caucasian descent (for the same reason as above). The multiple thresholds of 80 and 88 cm in women and 94 and 102 cm in men refer to cutoffs delineating successive levels of cardiovascular risk among patients with abdominal obesity and metabolic syndrome. The same remark for lines 125-126.
- Lines 77-80: raising the clinical suspicion of metabolic syndrome does not require measuring height and weight, although this is indeed useful for other purposes. Abdominal circumference and blood pressure are enough for the primary care physician to identify the metabolic syndrome-prone phenotype.
- Table 1: please realign items in the first column to correspond with the product types in the second.
- Line 120, the prevalence of excess weight among adults: I would advise the authors to consult (and refer to) the 2024 edition of the World Obesity Atlas.
- Line 212, the inferior cutoff value for fasting plasma glucose in prediabetes: even though American Standards of Care (which, by the way, upgraded to the 2025 edition, if the authors maintain the decision to quote them) are widely read and used, we should not forget that the WHO and IDF did not yet adopt the American definition of prediabetes. These international scientific forums still define impaired fasting glucose as starting from 110, not 100 mg/dL.
A series of minor typing and syntactic errors impair the background of an otherwise good-quality English language. The authors should revise the text once more to deal with the few omitted articles and commas, switched letters (see “diary” instead of “dairy” in line 239) and double-blank spaces.
Author Response

(The authors gave the same response as above.)

Reviewer 3 Report
Comments and Suggestions for Authors
The present review by Muszalska et al. examines the role of a rising dietary pattern, the planetary diet, in one of the most prevalent pathologies in Western societies: metabolic syndrome. While, in general, this is a well-written review that explores the topic in depth, I outline below a series of points that the authors should address to improve it. These mainly concern the use of terms and abbreviations, as well as some additional reflections for discussion.
1. The aim of the study should be explicitly stated in the abstract.
2. Line 46: The abbreviation MetS should be used here, as well as in lines 48 and 53. In line 53, consider starting the sentence with “The MetS” to avoid beginning with an abbreviation. Please review the entire manuscript to ensure consistency in the use of abbreviations.
3. Metabolic syndrome: I suggest expanding the section discussing the MetS criteria, including how these have evolved—initially incorporating more cardiovascular risk factors and later encompassing a broader range of cardiometabolic risk factors.
4. Lines 81–84: I recommend a deeper analysis of the factors contributing to MetS. While the role of lifestyle is crucial, it would be beneficial to also address the social determinants of health, given the well-documented adverse effects of low socioeconomic status and low educational attainment. Additionally, I pose the following question: Do people consume nutrients or foods? It might be more relevant to real-world applications to discuss dietary patterns in terms of food groups (e.g., ultra-processed foods) rather than individual nutrients (e.g., saturated fats). Ultra-processed foods are harmful not only due to their nutrient composition (high in calories, unhealthy fats, free sugars, salt, refined flours, and additives, and low in fiber) but also due to their accessibility, affordability, and displacement of healthier food options (such as fruits, vegetables, and nuts). This aligns with your introduction, where you highlight the convenience of these products in modern lifestyles.
5. Lines 128–129 & 177: The abbreviation CVD should be used. Please ensure abbreviation consistency throughout the manuscript.
6. Line 148: The term “planetary health diet (PHD)”—does this refer to the same concept as “planetary health”, or are there distinctions? If they are nearly identical, I suggest using a single term consistently to avoid confusion.
7. I recommend replacing “abovementioned” with “above-mentioned” for correctness and readability.
8. Lines 176–179: You reference the PREDIMED study, but a citation is missing. Additionally, I suggest reviewing the following PREDIMED-Plus studies, which provide a broader scope and relevant findings that could enrich your review:
DOI: 10.1007/s00394-021-02647-4 (Examines pro-vegetarian dietary patterns and cardiometabolic risk, showing positive effects on MetS score, BMI, and waist-to-hip ratio when the diet includes healthy plant-based foods).
DOI: 10.1016/j.atherosclerosis.2023.05.022 (Investigates ultra-processed food consumption, showing longitudinal associations with increased weight, BMI, and waist circumference).
9. Line 229: You should explicitly state that this association was non-significant (HR: 0.90; 95% CI: 0.75–1.10). Please ensure careful interpretation of results from other studies.
10. Lines 284–285: Regarding the mechanisms involved in the detrimental effects of meat consumption on diabetes, I suggest incorporating information on heme iron. This is a relatively new hypothesis with substantial epidemiological evidence (e.g., DOI: 10.1038/s42255-024-01109-5) and could enhance your discussion.
11. Lines 399–400: This is not the first mention of RCTs, so the abbreviation should be introduced earlier (line 161). Please ensure consistency in defining abbreviations upon their first appearance.
12. Line 400: The abbreviation PS should be used.
13. Line 488: Since this index was first mentioned in line 342, the abbreviation PDHI should be defined there for clarity.
14. Line 497: I suggest replacing “In the previous chapters” with “In the previous sections” for better clarity.
15 Terminology consistency: Throughout the manuscript, various terms are used to describe the central concept (planetary diet, planetary health diet, planetary health diet index, planetary health diet score), sometimes with capitalization and sometimes without. If these terms refer to the same concept, I strongly recommend standardizing the terminology to avoid confusion. Please review the manuscript for consistency.
16. Line 533: For consistency with previous abbreviatures, I suggest writing “The Healthy Reference Diet (HRD)” instead of placing the abbreviation first.
17. Finally, I recommend including a graphical abstract. This would serve as a valuable tool for visually summarizing your review and making it more engaging for a broader audience.
Author Response

(The authors gave the same response as above.)

Round 2
Reviewer 1 Report
Comments and Suggestions for Authors
Dear Authors,
missing conflicts:
- editing references aren't still suitable for the editor template;
- I suggest in the title use of : and not - "cardiovascular disease: a narrative review";
- Sanra check list missing in the supplementary file annex;
- native review suggest.
Native english request
Author Response
Dear Reviewer,
Thank you very much for your comments and suggestions. We do apologize for incorrect reference format. We made neccessary changes. The change in the title was introduced, SANRA checklist was uploaded as a supplementary material. The whole manuscript was proofread by a native speaker.
Kind regards,
Authors
Reviewer 2 Report
Comments and Suggestions for Authors
Compared to the previous version of the manuscript, the current form of the paper has definitely improved. I congratulate the authors for their substantial and serious work. However, I still disagree with the authors on two topics:
- Waist circumference (WC) thresholds:
I still think that normal WC thresholds refer to values of 80 cm in women and 94 cm in men of Caucasian descent (for the same reason as above). I do not deny the existence of multiple thresholds (80 and 88 cm in women and 94 and 102 cm in men) used by international guidelines; still, I maintain my opinion that they refer to cut-offs delineating successive levels of cardiovascular risk among patients with abdominal obesity and metabolic syndrome. My feedback on the second and third sources the authors quote is as follows:
- The 2019 Endocrine Society guideline focuses on the definition of metabolic risk as reflecting an individual’s predisposition for developing ASCVD and/or T2DM and not on the definition of metabolic syndrome. In Appendix A, where the guideline’s authors discuss the choice of terminology, they make the following assertion: “Furthermore, because these definitions do not contain all ASCVD risk factors and dichotomise the population into those with and without the metabolic syndrome, it should not be used as an indicator of absolute, short-term risk for ASCVD. The occurrence of multiple metabolic risk factors in one individual, nonetheless, does indicate the presence of a higher long-term risk for both ASCVD and T2DM.” In other words, they view the issues of metabolic syndrome vs metabolic risk as resembling, but not entirely overlapping, clinical entities. Therefore, their definition of metabolic risk should not be used to define WC limits delineating metabolic syndrome.
- The 2019 European guideline for adult obesity management in primary care “Normal waist circumference references (relatively strict values, even for certain non-obese and elderly individuals) are < 80 cm for women and < 94 cm for men. Cut-off points indicating higher cardiometabolic risks are > 88 cm for women and > 102 cm for men.” In other words, these authors also distinguish between values used to diagnose abdominal obesity by itself and the global cardiometabolic risk of an individual.
- The inferior cut-off value for fasting plasma glucose in prediabetes:
Being faced with different cut-offs depending on the organisation, the authors quote a so-called unified source, a 2023 review published in JAMA. However, as JAMA is an American journal, it is no wonder they use the American cut-offs. Far from being entirely wrong, this is a less representative choice for readers belonging to non-American countries, which may view this narrowed approach as a shortcoming of the manuscript.
Ultimately, I think these two matters belong to the commonplace circumstance of different interpretations of scientific sources by authors from various medical schools. Therefore, I leave the final decision at the editor’s and authors’ discretion. However, they should remember that they aim for an international audience as large as possible, and some readers may feel less represented by a narrowed-down assertion.
Author Response
Dear Reviewer,
Thank you very much for your suggestions. We held a debate regarding those various cut-offs and how to include those information in our manuscript. To make sure that we represent the international approach, we decided to underline the fact that there are various cut-offs, depending on the recommendations (both for WC and FPG). Therefore, we added the information about the prediabetes diagnosis. Those various criteria are described in JAMA review, so not including them was not right. Additionally, we updated the references with ADA 2025 guideline. Apologies, for missing that earlier.
Our manuscript is now also proofread by the native speaker. We hope that all of those changes improve the quality of our work.
Kind regards,
Authors
Reviewer 3 Report
Comments and Suggestions for Authors
The authors have addressed most of my comments. However, I still have a few minor comments:
- Line 217: There are two instances of "that"; one should be removed.
- Line 245: There is a bracket at the end of the sentence, after "adults." Please review and delete it.
- Lines 305-306: Should it be "diary" or "dairy"? Please review and correct if necessary.
- Line 363: "Diabetes" should be replaced with the abbreviation "T2D."
- Lines 363-364: One instance uses "heme iron," while the other uses "haem iron." Please unify for consistency.
- Line 629: "Diabetes" should be replaced with "T2D."
Author Response
Dear Reviewer,
Apologies for overlooking those evident errors. We introduced neccessary corrections. Moreover, the whole manusript was proofread by a native speaker.
Kind regards,
Authors